# Epigenetic Mechanisms of Inflammasome Regulation

**DOI:** 10.3390/ijms21165758

**Published:** 2020-08-11

**Authors:** Giulia Poli, Consuelo Fabi, Marina Maria Bellet, Claudio Costantini, Luisa Nunziangeli, Luigina Romani, Stefano Brancorsini

**Affiliations:** 1Department of Experimental Medicine, University of Perugia, 06132 Perugia, Italy; marinamaria.bellet@unipg.it (M.M.B.); costacla76@gmail.com (C.C.); luigina.romani@unipg.it (L.R.); stefano.brancorsini@unipg.it (S.B.); 2Department of Surgical and Biomedical Sciences, Urology and Andrology Clinic, University of Perugia, 05100 Terni, Italy; consuelofabi93@gmail.com; 3Polo d’Innovazione di Genomica, Genetica e Biologia, 05100 Terni, Italy; luisa.nunziangeli@gmail.com

**Keywords:** inflammasome, microRNA, epigenetic modifications

## Abstract

The innate immune system represents the host’s first-line defense against pathogens, dead cells or environmental factors. One of the most important inflammatory pathways is represented by the activation of the NOD-like receptor (NLR) protein family. Some NLRs induce the assembly of large caspase-1-activating complexes called inflammasomes. Different types of inflammasomes have been identified that can respond to distinct bacterial, viral or fungal infections; sterile cell damage or other stressors, such as metabolic imbalances. Epigenetic regulation has been recently suggested to provide a complementary mechanism to control inflammasome activity. This regulation can be exerted through at least three main mechanisms, including CpG DNA methylation, histones post-translational modifications and noncoding RNA expression. The repression or promotion of expression of different inflammasomes (NLRP1, NLRP2, NLRP3, NLRP4, NLRP6, NLRP7, NLRP12 and AIM2) through epigenetic mechanisms determines the development of pathologies with variable severity. For example, our team recently explored the role of microRNAs (miRNAs) targeting and modulating the components of the inflammasome as potential biomarkers in bladder cancer and during therapy. This suggests that the epigenetic control of inflammasome-related genes could represent a potential target for further investigations of molecular mechanisms regulating inflammatory pathways.

## 1. Introduction

Inflammasomes are cytoplasmic multiprotein complexes sensing microbial pathogens or detecting sterile danger and mediating the activation of proinflammatory cytokines in the context of the innate immune response [1]. It is acknowledged that the basal levels of the inflammasome components are low to prevent inappropriate activation and that two distinct steps are required for the assembly and function of the inflammasome: a first priming step upregulates the levels of inflammasome components, and a second activation step ignites the oligomerization of inflammasome complex [2]. Among the transcriptional and post-transcriptional mechanisms that control the levels of inflammasome components in basal and activating conditions [3,4], accumulating evidence indicates that epigenetic events may actually play a critical role, the failure of which results in the development of pathological conditions, characterized by either hyper- or hypoactivation of the inflammasome. In this review, starting from the definition of the most important epigenetic modifications and their significance, we summarize evidence suggesting their critical role in the regulation of inflammasome activity and inflammasome-related diseases.

## 2. Epigenetic Modifications

The regulation of biological processes can be achieved through genetic and epigenetic mechanisms. Epigenetic regulation is traditionally defined as a mechanism generating “potentially heritable changes in gene expression not involving changes in DNA sequence”. These changes can be exerted through at least three main mechanisms, including CpG DNA methylation, histones post-translational modifications (PTMs) and noncoding RNA expression [5]. All of these events contribute together to the significant variations in cellular functions within a single organism.

The first mechanism, DNA methylation, occurs upon the addition of a methyl group at the fifth position of the pyrimidine ring of the cytosine residues at the level of CpG dinucleotides and functionally results in transcriptional repression [6]. DNA methylation has been firmly established as a crucial epigenetic mechanism involved in a number of key processes, including development, aging and carcinogenesis.

Histones represent the proteins around which DNA is wrapped to form the nucleosomes, basic units of the chromatin, and are substrates for a wide range of PTMs occurring at multiple amino acid residues within their N-terminal tails, including phosphorylation, acetylation, methylation, sumoylation, ubiquitylation, ADP-ribosylation, biotinylation and the recently identified serotonylation [7]. Therefore, a large variety of chromatin remodeling enzymes interact with chromatin and are implicated in adding, reading or removing PTMs, collectively influencing the chromatin structure and defining the accessibility of the chromatin to transcriptional expression, thus generating the high level of plasticity of chromatin remodeling [8]. Therefore, alteration in the activity of these enzymes can often result in pathologic conditions [9].

Noncoding RNAs are untranslated transcripts that have important regulatory roles in cellular biology. They can be classified in short, mid and long based on their length [10]. Among noncoding RNAs, microRNAs and long noncoding RNAs (lncRNAs) are the most extensively studied in several pathologies. LncRNAs regulate gene expression by interacting with DNA, messenger RNAs (mRNAs) and proteins, while miRNAs mediate the post-transcriptional repression or mRNA degradation according to epigenetic mechanisms [11,12]. miRNAs belong to short noncoding RNAs with 22–25 nucleotides in length, and their activity is performed with peculiar complementarity miRNA-mRNAs [12].

## 3. Inflammasomes and Inflammasome-Associated Diseases

The scientific evidence of “inflammasome” existence dates back to 2002, when a caspase-activating complex composed of the NLR (nucleotide-binding oligomerization domain, Nod-like receptor) protein, the adaptor ASC (apoptosis-associated speck-like protein containing a CARD) and pro-caspase-1 was identified [13]. Several types of inflammasomes have been identified that can respond to distinct bacterial, viral or fungal infections, cell damage and other stressors, such as metabolic imbalances. Inflammasomes are composed of a sensor, such as pyrin domain containing related protein family (NLRP), absent in melanoma (AIM) 2, CARD domain containing (NLRC) 4, the adaptor ASC and caspase-1. While NLRP1, NLRP3, NLRP6, NLRP7, NLRP12 and AIM 2 depend critically on ASC for the engagement of caspase-1, the inflammasome sensor NLRC4 can directly interact with caspase-1. The regulated activation of inflammasomes after microbial infection or injury is critical for the maintenance of tissue homeostasis, and deregulated inflammasome activity has emerged as a major contributor to the pathogenesis of prevalent diseases, including inflammatory bowel disease, coronary heart disease, autoimmune diseases, neurodegenerative diseases and cancer [14]. Together with Toll-like receptors (TLRs) [15], NLRs belong to the innate immune pattern recognition receptors (PRRs), which are responsible for the recognition of pathogens or tissue injuries through pathogen-associated molecular patterns (PAMPs) and danger-associated molecular patterns (DAMPs), respectively [16]. The culminant event is the production of active caspase-1 able to process prointerleukin-1β (IL-1β) into the mature protein IL-1β. Progressively, inflammasomes have attracted scientific interest, and nowadays, they are considered crucial mediators of inflammation. Currently, NLRPs, AIM2, IFI (interferon-γ inducible factor) 16 and RIG-I inflammasomes have been characterized [17]. The NLRP3 inflammasome is the most studied one. Its role has been characterized first in a subset of rare autoinflammatory conditions [18] and, subsequently, in inflammatory [19,20,21,22,23] and metabolic diseases [24]. In contrast, the mechanism of action of other inflammasomes such as NLRP7 [25] and IFI16 [26], have been more recently described. In particular, NLRP7 has been shown to assemble an ASC and caspase-1-containing high molecular weight inflammasome complex in response to bacterial lipopeptides, with a molecular mechanism involving ATP binding and direct ATPase activity [25]. Instead, IFI16 [26] and AIM2 [27] are involved in inducing a caspase-1-activating inflammasome formation by recognizing viral DNA in the nucleus or cytosol, respectively. A recent study identified a mechanism by which AIM2 inflammasome activity is negatively regulated by TRIM11 (tripartite motif 11), determining its degradation via selective autophagy upon viral infection [28]. Recently, novel significantly divergent functions for NLRPs have been identified, in addition to their already well-established proinflammatory functions. This is especially true for NLRP12 and NLRP4. NLRP12 was first described 10 years ago, when a seminal study demonstrated its role as a negative regulator of the nuclear factor kappa-light-chain-enhancer of activated B cells (NF-κB) signaling pathway [29]. NLRP12 plays a crucial role in both the hematopoietic and nonhematopoietic compartment for controlling overt inflammation, colitis and colitis-associated tumorigenesis. The absence of NLRP12 in mice resulted in a severe uncontrolled inflammation that rendered NLRP12-deficient mice highly susceptible to experimental colitis and inflammation-induced tumorigenesis [29]. NLRP4 (a member of the NLR family of the cytosolic receptor strongly expressed in several tissues [30]) has recently been reported as a negative regulator of autophagy and type I IFN signaling, resulting from the interaction of its NACHT domain with Beclin1 and TANK-binding kinase 1 (TBK1), respectively [31]. Furthermore, NLRP4 was identified as an inhibitor of tumor necrosis factor (TNF)-α- and IL-1β-mediated NF-κB activation, which is achieved through an interaction with IKKα. The pyrin domain (PYD) of NLRP4 is necessary for this inhibitory effect on NF-κB, underscoring its importance as a critical regulator of inflammatory signaling pathways [32]. Moreover, recent literature frames NLRP4 as an important key mediator during the immune response to viral infections [33].

During the last decades, there has been an explosion in the research field of inflammasomes; the majority of research articles were aimed at the characterization of inflammasomes in noncancerous diseases, ranging from autoimmune diseases, such as arthritis [34,35,36] and lupus [37,38], to neurodegenerative [39,40], renal [41,42,43] and vascular diseases [44,45]. The main efforts at dissecting inflammasome mechanisms of action came from studies using murine or cell models of fungal [46,47,48,49,50], bacterial [51,52,53,54,55,56], viral [57,58,59,60,61] and prion infections [62]. On the one hand, by using animal and cell models, the function of each unit of the inflammasome apparatus can be assessed manipulating its expression. On the other hand, genetic studies uncovering mutations or genetic variants in inflammasome components, conferring the susceptibility for diseases such as diabetes and systemic inflammation [63], provided the complementary side of this intrinsically complex picture. A recent review highlighted the importance of inflammasome-related genes in several major human diseases, by molecular and genetic networks, based on genome-wide association studies [64]. Increasingly available large Omics and clinical data, in tandem with system biology approaches, have offered the opportunity to study more comprehensive and dynamic molecular inflammation networks, showing a double-sword role of inflammasomes. A recent study identified 12 expression profiling datasets derived from nine different tissues isolated from 11 rodent inflammatory disease models related to common chronic diseases. The overlapping of inflammasomes with the innate immunity genes generated a list of six common complex diseases, including obesity, type II diabetes, coronary heart disease, late-onset Alzheimer’s disease, Parkinson’s disease and sporadic cancer [65]. The crucial crosslink between inflammasomes and immune-related diseases is the activation of IL-1β; immune stromal and tumor cells can produce IL-1β, which also stimulates the expression of cyclooxygenase (COX)-2, IL-6 and chemokine C-C motif ligand (CCL) [66].

Although the importance of IL-1β in cancer is without doubt, the list of its cellular targets is still partially defined. T lymphocytes and myeloid cells are established downstream targets of IL-1β; in myeloid cells, IL-1β activates the NF-κB pathway through binding to its receptor, IL-1RI [67]. The ability of IL-1β in inducing angiogenic pathways, which trigger tumor progression, has been widely reported [68], although the mechanisms through which this event occurs have not been completely defined. Due to the relative novelty of this topic, scientific reports dissecting the role of inflammasomes in carcinogenesis are still a minority, compared to infectious and autoimmune diseases. In fact, a plethora of scientific works have been aimed at the characterization of inflammasomes in melanoma [69], leukemia [70] colon [71,72], oral [73] and colitis-associated [74] cancers. Products of inflammasome activation (IL-1β and IL-18) behave as protumorigenic factors in gastrointestinal cancers, while the protective role of NLRP6 against tumor development has been clarified [75], whereas, for NLRC4, contrasting findings were reported [76,77]. A constitutive activation of the NLRP3 inflammasome in late-stage human melanoma cells with the autonomous secretion of active IL-1β has been demonstrated [78]. Knockout mice for IL-1β have reduced the angiogenesis and growth of melanoma tumors [79]. The treatment of metastatic human and mouse melanoma cell lines with the anti-inflammatory phytochemical thymoquinone was shown to hamper the metastasis process by inhibition of the NLRP3 inflammasome [80]. Nonetheless, in hepatocellular carcinoma, the loss in expression of the NLRP3 inflammasome components, both mRNA and protein, was demonstrated and correlated with cancer progression [81].

## 4. Epigenetic Regulation of the Inflammasome

A role for dysfunctional epigenetic mechanisms has been investigated in autoinflammatory diseases, pathological conditions characterized by recurrent episodes of systemic inflammation without infection and autoimmunity [82]. Several genes have now been associated with monogenic autoinflammatory syndromes [83]. For instance, Cryopyrin-associated periodic syndromes (CAPS) are caused by gain-of-function mutations in the *NLRP3* gene that result in the constitutive activation of the NLRP3 inflammasome and the release of IL-1β, in turn responsible for the symptomatology of the disease [84]. It is clear, however, that the clinical manifestations of autoinflammatory syndromes are variable, and environmental and genetic factors, including epigenetic dysregulation, may contribute to the disease presentation and outcome [85,86]. Prompted by the observation that inflammasome-related genes were demethylated during macrophage differentiation and monocyte activation, Vento-Tormo and co-authors analyzed a cohort of patients with CAPS for eventual epigenetic dysregulation [87]. While unstimulated monocytes had similar levels of DNA demethylation in patients and healthy controls, the IL-1β stimulation resulted in a higher demethylation of inflammasome-related genes such as *IL1B*, *IL1RN*, *NLRC5* and *PYCARD* in CAPS patients [87]. Of note, the lower methylation levels were observed only in CAPS patients who did not receive the anti-IL-1 treatment, suggesting that IL-1 drives the different methylation patterns [87]. In agreement with these findings, monocytes from patients with chronic nonbacterial osteomyelitis/chronic recurrent multifocal osteomyelitis, an autoinflammatory bone disorder, had reduced methylation levels in the *NLRP3* and *PYCARD* genes compared to healthy controls [88]. Therefore, dysregulated epigenetic mechanisms may contribute to the clinical manifestations of autoinflammatory syndromes by upregulating the expression of inflammasome components. However, this is not a general mechanism, as the same study of Vento-Tormo and co-authors could not identify differences in the methylation patterns between patients with familial Mediterranean fever (FMF), an autoinflammatory disease caused by mutations in the gene *MEFV* encoding for the pyrin protein, and healthy controls [87]. The low number of patients may have hindered the detection of a statistically significant difference. Alternatively, since lipopolysaccharide (LPS)-stimulated monocytes from FMF patients secrete higher amounts of IL-1β compared to healthy controls in a NLRP3-dependent manner [89], other mechanisms may play a major role in the dysregulation of inflammasome activity.

The epigenetic regulation of the inflammasome has also been investigated in infectious diseases. For instance, the challenge of TPH-1 with *Mycobacterium tuberculosis* resulted in the demethylation of the *NLRP3* promoter, suggesting that epigenetic mechanisms may contribute to the activation of the inflammasome during infection [90]. In other cases, however, the pathogen co-opts endogenous mechanisms of the epigenetic regulation to downmodulate the inflammasome activation and promote pathogen colonization, as recently demonstrated for *Leishmania amazonensis* [91]. Indeed, the parasite was able to remodel the chromatin by reducing the H3K9/K14 acetylation and H3K4 trimethylation, resulting in the inhibition of the NF-κB pathway and prevention of NLRP3 activation [91]. Therefore, epigenetic regulation represents a double-edged sword in infectious diseases; on the one hand, it allows inflammasome priming and the activation for a proper response to infections, but, on the other hand, it can be exploited by pathogens to protect them selves from the immune response.

The epigenetic regulation of the inflammasome may also be co-opted in tumors. Indeed, the adaptor protein ASC, also known as target of methylation-induced silencing-1 (TMS1), is a sensitive target of DNA methylation in different tumors, and its demethylation sensitizes cancer cells to apoptosis [92]. In addition, the downregulation of ASC prevents inflammasome assembly and activation [92]. However, while the function of ASC as a tumor-suppressor protein in different types of cancers is acknowledged, the role of inflammasomes is disputed, and pro- and antitumoral effects have been described [93,94]. Therefore, the specific context of each tumor should be carefully evaluated to decide whether ASC de-repression and inflammasome activation would represent a beneficial strategy [95]. The epigenetic dysregulation of the inflammasomes has also been associated with the effects of chemotherapic drugs. For instance, bortezomib, a proteasome inhibitor, promoted histone H3 and H4 acetylation on the *NLRP3* promoter via STAT3, resulting in the upregulation of *NLRP3* expression in dorsal root ganglion, a potential mechanism for bortezomib-induced painful neuropathy [96]. On the contrary, low doses of epirubicin, an anthracycline drug, displayed anti-inflammatory properties by downregulating the NLRP3 inflammasome, and this was associated with a reduction of histone 3 lysine 9 acetylation [97].

Other pathologies have been linked to the dysregulation of epigenetic mechanisms in the control of inflammasome expression. For instance, in the peripheral blood of juvenile spondylarthritis patients, a higher methylation of the *NLRP3* gene promoter was observed, in-line with a decreased expression of the protein in these patients, and a vicious cycle of microbiota dysbiosis and reduced NLRP3 inflammasome has been hypothesized to take place in the disease [98]. Conversely, an increased histone acetylation at the *NLRP3* promoter and NF-kB activation contributes to the activation of the NLRP3 inflammasome in vascular smooth muscle cells and consequent phenotypic changes and proliferation in hypertension [99]. Moreover, treatments with histone deacetylase (HDAC) 3 inhibitors have been shown to alleviate the inflammatory response and protect against ischemic brain damage by downregulating the AIM2 inflammasome [100]. Similarly, the pharmacological inhibition of the histone demethylase Jumonji domain-containing 3 (Jmjd3) has been associated with reduced inflammation by limiting NLRP3 activation in a murine model of DSS-induced colitis [101].

All in all, the available evidence supports the notion that DNA methylation and histone modifications play a critical role in the regulation of the expression of the inflammasome components, and dysregulation may increase the susceptibility to pathological conditions. It comes that the pharmacological targeting of epigenetic mechanisms might represent a valuable strategy to restore the homeostatic regulation of the inflammasome and balance between the need for inflammasome function in response to environmental cues and the prevention of tissue damage by unrestrained activation.

## 5. Role of Noncoding RNA in the Regulation of Inflammasome

Recent studies demonstrated the post-transcriptional regulation of inflammasomes exerted by miRNA. They can repress or increase inflammasome gene expressions by binding to their mRNA target. In the literature, different miRNAs are known to regulate different inflammasomes. NLRP3 represents the most-studied inflammasome, and some validated miRNAs that exert a post-transcriptional regulation on this are well known. Table 1 shows inflammasome-validated miRNAs. 

In 2012, Bauernfeind et al. identified miR-233-3p as the first human miRNA that directly regulates NLRP3. MiR-223 is highly expressed in the myeloid cell lineage, especially neutrophils, and is absent in B cells and T cells, contrary to NLRP3, which results are down-expressed [118]. Moreover, the miR 223-3p decreases and the NLRP3 increases in monocyte differentiation, suggesting that this small noncoding RNA exerts a possible role during this differentiation phase [102]. *NLRP3* presents a conserved binding site for miR 223-3p in its 3′ UTR region. The binding between this conserved region and miR 223-3p causes a reduction of NLRP3 activity. Studies in vitro revealed that the *NLRP3* target region is conserved among mammals; a mutation in this region causes a complete loss of the miR-223–mediated regulation of the *NLRP3* 3′ UTR [118]. Bandyopadhyay et al. investigated the involvement of miR-133a-1 in inflammasome activation (*NLRP3*) and IL-1β production [105]. In this work, they overexpressed and suppressed miR-133a-1 in differentiated THP1 cells expressing NLRP3. The overexpression of miR-133a-1 increased caspase-1 and IL-1β levels in response to inflammasome stimuli. The results demonstrated that miR-133a-1 does not alter the basal expression of individual components of the NLRP3 complex; however, it regulates IL-1β processing and expression levels of its effector protein caspase-1 [105]. The role of NLRP3 activity was also analyzed in gastric cancer. NLRP3 is markedly upregulated in gastric cancer, which promotes NLRP3 inflammasome activation and IL-1β secretion in macrophages [106]. Li et al. identified miR-22 as constitutively expressed in gastric mucosa, where it directly targets NLRP3 at the transcript level [119]. This binding attenuates NLRP3 oncogenic effects in vitro and in vivo. Moreover, NLRP3 inflammasome can be activated by *Helicobacter pylori* (H. pylori) infection resulting in IL-1β secretion [119]. The H. pylori infection markedly suppresses miR-22 expression, an event that prevents miR-22 from suppressing NLRP3 expression, attenuating NLRP3-driven cell proliferation and preventing gastric cancer carcinogenesis [106,120]. 

Other miRNAs can influence the NLRP3 inflammasome, such as miR-21, which is expressed aberrantly and plays a role in LPS-induced septic shock [121]. A recent study investigated the role of this miRNA and demonstrated that miR-21 deficiency inhibited NLRP3, ASC and caspase-1 expressions. They found that knockout mice for miR-21 presented inhibited caspase-1 activation and IL-1β secretion [122]. Other studies elucidated the role of miR30e in inflammation. This miRNA is a key inflammation-mediated molecule that could be a potential target for NLRP3 [107]. Li et al. found that *NLRP3* shows conserved miR-30e binding sites in its 3′UTR, suggesting a link between miR-30e and NLRP3 inflammasome-mediated neuroinflammation in the pathogenesis of Parkinson’s disease (PD). There is evidence that miR-30e improves neuronal damage via negatively regulating NLRP3 expression and inhibiting NLRP3 inflammasome activation in induced PD mice models. The post-transcriptional regulation of NLRP3 mRNA and protein expression is negatively regulated by miR30e [107], confirming its critical role in PD pathogenesis [123]. Junn et al. provided a direct link between miR-7 and NLRP3 inflammasome-mediated neuroinflammation in PD pathogenesis. Specifically, it has been known that miR-7 directly regulates α-Syn expression in dopaminergic neurons via post-transcriptional regulation and is associated with the pathophysiology of PD [123]. Inflammasomes containing NLRP3 are highly expressed in microglia, and they are essential to the process of neuroinflammation [124]. Activated microglia produce a large number of inflammatory cytokines, contributing to dopaminergic neuronal degeneration [108]. NLRP1 inflammasome, the first inflammasome to be discovered and characterized [17], exhibits post-transcriptional regulation exerted by miRNA. Recent studies demonstrated that NLRP1 is regulated by miR-199a-3p and is significantly downregulated in acute lung injury (ALI) tissues [109]. A microarray analysis revealed the presence of miR-199a-3p in three of the 24 lung tissues collected from donor patients who died with ALI. Moreover, miR-199a-3p downregulation eliminates the inhibition of NLRP1, causing the activation of NLRP1 and pro-IL-1β and pro-IL-18 cleavage mediated by caspase-1. Consequently, high levels of IL-1β and IL-18 are produced, further exacerbating the inflammatory response and causing the recurrence of the disease [109]. Recently, an epigenetic role of miRNAs targeting other inflammasome genes has been demonstrated. A recent study identified a conserved binding site for miR-372 in the 3′UTR of *NLRP12* and demonstrated that miR-372 is particularly present in the blood and colonic mucosa of ulcerative colitis patients compared with healthy controls. Here, the overexpression of miR-372 significantly decreased the protein expression level of NLRP12, thus inducing excessive inflammatory cytokine production and disease progression [115]. In 2013, Momeni and colleagues demonstrated that the miR-143 transfection in Jurkat cell lines determined the increased level of AIM2 transcript suggesting a possible role of miR-143 in the targeting of AIM2 [116]. Recently, a potential regulatory role of miR-18b on NLRP7 has been shown. In particular, miR-18b was shown to bind a 3′-UTR specific binding site in the *NLRP7* gene in a breast cancer cell line, where miR-18b was found upregulated. A miR-18b down-expression in these cells induced an upregulation of NLRP7 that promoted cell migration and metastasis [117].

Finally, it is worth mentioning that our group has recently explored the role of miRNAs targeting and modulating inflammasome components as potential biomarkers in bladder cancer. Our studies demonstrated the variable expression of different NLRP genes, including *NLRP3*, *NLRP4*, *NLRP9* and *NAIP*, and of miRNAs targeting these NLRs (miR-146a-5p, miR-106a-5p, miR-17-5p, miR-223-3p, miR-141-3p, miR-19a-3p, miR-145-5p and miR-185-5p) in the urine sediments of patients harboring bladder cancer compared to the control healthy subjects. Specific correlations with the tumor stage, risk of recurrence and response to intravesical Bacillus Calmette-Guérin instillation were identified, thus suggesting that the assessment of the expression level of inflammasome-related genes and their regulatory miRNA could represent a potential reliable noninvasive tool for diagnosis in patients with bladder cancer, as well as predictive markers of the responses to therapy [114,125,126]. 

## 6. Conclusions

Inflammasomes are important regulators of inflammation. A dysregulation of their expression can cause the onset of different pathologies. Recent studies have identified the role played by epigenetic mechanisms in the modulation of inflammasomes expression. DNA methylation, histone modifications and miRNA inhibitory activity, key players of epigenetic regulation, can act by modulating the expression of inflammasome components. The dysregulation of these mechanisms may increase the susceptibility to pathological conditions (Figure 1), suggesting a potential role of epigenetic factors as pharmacological targets to restore the homeostatic regulation of the inflammasome.

Finally, epigenetic modifications could have valuable importance as pathological predictive biomarkers in biological fluids, as well as in tissues, and the interpretation of epigenetic unbalance could be decisive for the diagnosis of diseases in clinical activity. Although successfully identified in basic research, the knowledge of epigenetic mechanisms is still rarely applied in clinical uses. In the future, clinical studies confirming the pivotal role of epigenetic modulators in diseases such as cancer or neurodegeneration will demonstrate their importance in diagnosis and treatment.

## Figures and Tables

**Figure 1 ijms-21-05758-f001:**
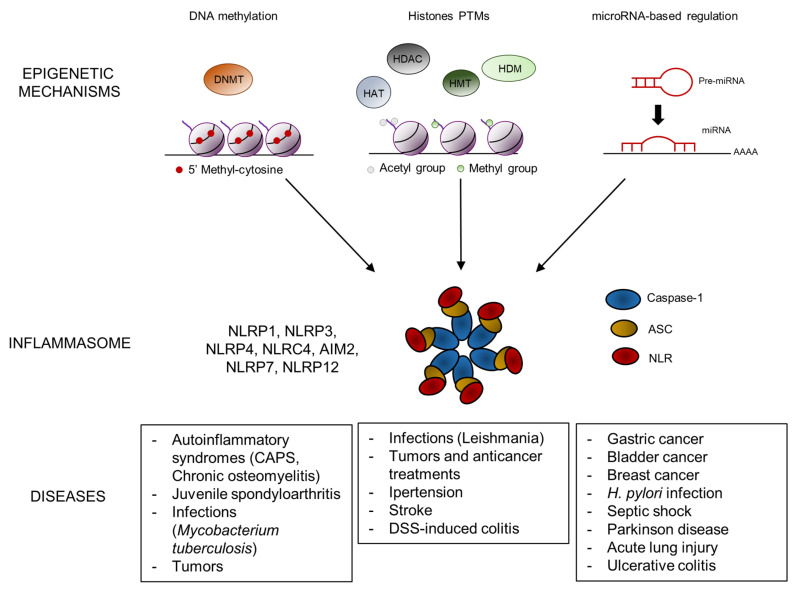
Schematic representation of the main epigenetic mechanisms involved in inflammasome activation. DNA methylation, histones post-translational modifications (PTMs) and microRNA-based regulation exert epigenetic control on the level of inflammasome activity, whose dysregulation may result in the development of inflammasome-related diseases. DNMT, DNA methyltransferases; HAT, histone deacetylases; HDAC, histone deacetylases; HMT, histone methyltransferases; HDM, histone demethylases; ASC, apoptosis-associated speck-like protein containing a CARD; NLR, NOD-like receptor and CAPS, cryopyrin-associated periodic syndromes; DSS, dextran sulfate sodium.

**Table 1 ijms-21-05758-t001:** List of miRNAs known to target inflammasome genes and associated diseases. NLR, NOD-like receptor.

miRNA	Target Gene	Disease	References
miR-223	*NLRP3*	Inflammatory bowel diseases	[102]
		Acute lung injury/acute respiratory distress syndrome	[103]
		Hepatocellular carcinoma	[104]
miR-133a	*NLRP3*	Inflammatory diseases	[105]
miR-22	*NLRP3*	Gastric cancer	[106]
miR-30e	*NLRP3*	Parkinson’s disease	[107]
miR-7	*NLRP3*	Parkinson’s disease	[108]
miR-199a-3p	*NLRP1*	Acute Lung Injury	[109]
miR-146a-5p	*NLRP3*	Autoimmune diseases, Multiple sclerosis	[110]
miR-20b-5p	*NLRP3*	Multiple sclerosis	[4]
miR-495-3p	*NLRP3*	Cardiac injury	[111]
miR-330-3p	*NLRP3*	Renal inflammatory disease	[112]
mir-17-5p	*NLRP3*	Obesity disease	[113]
mir-141-3p	*NLRP3; NLRP4*	Bladder cancer	[114]
miR-372	*NLRP12*	Ulcerative Colitis	[115]
miR-143	*AIM2*	Inflammatory diseases	[116]
miR-18b	*NLRP7*	Breast cancer	[117]

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
