# Peer review of "Epigenetic Mechanisms of Inflammasome Regulation"

_ijms, 2020, doi:10.3390/ijms21165758_

Round 1

Reviewer 1 Report

The review has presented literature on the epigenetic mechanisms of inflammasome regulation. Whilst this concept is covered in detail, there appears to be a prime focus on the regulation of the NLRP3 inflammasome. Consideration of how other inflammasomes NLRP1, NLRP2, NLRP3, NLRP4, NLRP7, NLRP12 and AIM2 would significantly add rounded to the current review.

Author Response

We thank the reviewer for his comment. Some paragraphs, table 1 and figure 1 have been modified according to reviewer suggestions. Both sentences and references added were marked in red color for easy reading.

Page 3, paragraph 3 titled "Inflammasomes and inflammasome-associated diseases", from line 98 to line 120

In particular, NLRP7 has been shown to assemble an ASC and caspase-1-containing high molecular weight inflammasome complex in response to bacterial lipopeptides, with a molecular mechanism involving ATP binding and direct ATPase activity [25]. Instead, IFI16 [26] and AIM2 [27] are involved in inducing caspase-1-activating inflammasome formation by recognizing viral DNA in the nucleus or cytosol, respectively. A recent study identified a mechanism by which AIM2 inflammasome activity is negatively regulated by TRIM11 (tripartite motif 11), determining its degradation via selective autophagy, upon viral infection [28]. Recently, novel significantly divergent functions for NLRPs have been identified, in addition to their already well-established pro-inflammatory functions. This is especially true for NLRP12 and NLRP4. NLRP12 was first described 10 years ago, when a seminal study demonstrated its role as a negative regulator of the NF-κB signaling pathway [29]. NLRP12 plays a crucial role in both the hematopoietic and non-hematopoietic compartment for controlling overt inflammation, colitis and colitis-associated tumorigenesis. The absence of NLRP12 in mice resulted in severe uncontrolled inflammation that rendered NLRP12- deficient mice highly susceptible to experimental colitis and inflammation-induced tumorigenesis [29]. NLRP4 (a member of the NLR family of cytosolic receptor strongly expressed in several tissues [30], has recently been reported as a negative regulator of autophagy and type I IFNs signaling, resulting from the interaction of its NACHT domain with Beclin1 and TANK-binding kinase 1 (TBK1), respectively [31]. Furthermore, NLRP4 was identified as an inhibitor of TNF-α- and IL-1β mediated NF-κB activation, which is achieved through an interaction with IKKα. The PYD of NLRP4 is necessary for this inhibitory effect on NF-κB, underscoring its importance as a critical regulator of inflammatory signaling pathways [32]. Moreover, recent literature frames NLRP4 as an important key mediator during the immune response to viral infections [33].

Page 8, paragraph 5 titled "Role of non-coding RNA in the regulation of inflammasome", from line 294 to line 305

Recently an epigenetic role of miRNAs targeting other inflammasome genes has been demonstrated. A recent study identified a conserved binding site for miR-372 in the 3′UTR of NLRP12 and demonstrated that miR-372 is particularly present in blood and colonic mucosa of ulcerative colitis patients compared with healthy controls. Here, overexpression of miR-372 significantly decreased the protein expression level of NLRP12, thus inducing excessive inflammatory cytokines production and disease progression [115]. In 2013, Momeni and colleagues demonstrated that miR-143 transfection in Jurkat cell lines determined increased level of AIM2 transcript, a result suggesting a possible role of miR-143 in the targeting of AIM2 [116]. Recently, a potential regulatory role of miR-18b on NLRP7 has been shown. In particular, miR-18b was shown to bind a 3’-UTR specific binding site in NLRP7 gene in breast cancer cell line, where miR-18b was found upregulated. A miR-18b down-expression in these cells induced an upregulation of NLRP7 that promoted cell migration and metastasis [117].

Page 6, table 1: miR-372, miR-143 and miR-18b were added

Page 9, figure 1: NLRP7 and NLRP12 were added

Reviewer 2 Report

Thanks for your comprehensive summary of the inflammasome regulation. It is a great job to summarize that CpG DNA methylation, histone post-translational modifications, and non-coding RNA expression are important for inflammasome regulation. Here are some small corrections.

  1. Line 13, "One of the most important inflammatory pathway..." should be " One of the most important inflammatory pathways...".
  2. Line 26, "inflammasome-related genes could represents..." should be "inflammasome-related genes could represent...".
  3. Line 39, "a failure of which result in the..." should be"a failure of which results in the..." .
  4. Line 42, "we summarize evidences suggesting..." should be "we summarize evidence suggesting...".
  5. Line 50, "contributes together..." should be "contribute together..."
  6. Line 189, "bortezomid" should be "bortezomib".
  7. Line 228, "non-coding RNA exert a possible..." should be "non-coding RNA exerts a possible... ".
  8. Line 255, "There are evidences that..." should be "There are evidence that... " 
  9. Line 279, "recurrence and response to intravescical..." should be "recurrence and response to intravesical ".
  10. Line 294, "microRNA-based regulation exhert..." should be " microRNA-based regulation exert...".
  11. Line 295, " whose disregulation may result... " should be " whose dysregulation may result ".

Author Response

We thank the reviewer for his corrections. We have modified the errors mentioned above and the entire review has been corrected. Right words were marked in red color for easy reading.

pathways: page 1, line 13

represent: page 1, line 26

results: page 1, line 39

evidence: page 1, line 42

contribute: page 2, line 50

bortezomib: page 5, line 211 and line 213

exerts: page 7, line 250

evidence: page 7, line 277

intravesical: page 8, line 312

exert: page 9, line 328

dysregulation: page 9, line 329

Round 2

Reviewer 1 Report

Thank-you for addressing the comments raised.